# MOF-Derived CoSe_2_@NiFeOOH Arrays for Efficient Oxygen Evolution Reaction

**DOI:** 10.3390/nano13192621

**Published:** 2023-09-22

**Authors:** Yulong Tang, Jiangning Li, Zhiyi Lu, Yunan Wang, Kai Tao, Yichao Lin

**Affiliations:** 1School of Materials Science & Chemical Engineering, Ningbo University, Ningbo 315211, China; tangyulong@nimte.ac.cn (Y.T.); lijiangning@nimte.ac.cn (J.L.); 2Key Laboratory of Advanced Fuel Cells and Electrolyzers Technology of Zhejiang Province, Ningbo Institute of Materials Technology and Engineering, Chinese Academy of Sciences, Ningbo 315201, China; luzhiyi@nimte.ac.cn; 3University of Chinese Academy of Sciences, Beijing 100049, China

**Keywords:** oxygen evolution reaction, NiFeOOH, water electrolysis, selenidation, CoSe_2_

## Abstract

Water electrolysis is a compelling method for the production of environmentally friendly hydrogen, minimizing carbon emissions. The electrolysis of water heavily relies on an effective and steady oxygen evolution reaction (OER) taking place at the anode. Herein, we introduce a highly promising catalyst for OER called CoSe_2_@NiFeOOH arrays, which are supported on nickel foam. This catalyst, referred to as CoSe_2_@NiFeOOH/NF, is fabricated through a two-step process involving the selenidation of a Co-based porous metal organic framework and subsequent electrochemical deposition on nickel foam. The CoSe_2_@NiFeOOH/NF catalyst demonstrates outstanding activity for the OER in an alkaline electrolyte. It exhibits a low overpotential (η) of 254 mV at 100 mA cm^−2^, a small Tafel slope of 73 mV dec^−1^, and excellent high stability. The good performance of CoSe_2_@NiFeOOH/NF can be attributed to the combination of the high conductivity of the inner layer and the synergistic effect between CoSe_2_ and NiFeOOH. This study offers an effective method for the fabrication of highly efficient catalysts for an OER.

## 1. Introduction

The storage of renewable energy presents a significant challenge to constructing an environmentally friendly energy system [1,2,3,4,5,6]. Water electrolysis has become a promising technology for storing electricity generated from renewable sources by converting it into a clean energy carrier, H_2_. It involves two distinct half-reactions: an hydrogen evolution reaction (HER) occurring at the cathode and an oxygen evolution reaction (OER) taking place at the anode [7,8,9,10]. Relative to the HER, the OER suffers from much more sluggish kinetics, and so, it requires highly efficient electrocatalysts. Thus far, Ir- and Ru-based materials represent state-of-the-art OER electrocatalysts because of their high activity level and stability [11,12,13,14,15,16,17]. However, the high cost and scarcity of these resources limit their extensive utilization [12,18,19,20]. Therefore, it is crucial to develop alternative, affordable electrocatalysts that are abundant in the Earth’s crust. Transition metal chalcogenides (TMCs) with the formula M_x_C_y_ (M = Fe, Co, and Ni; C = S and Se) have gained significant interest for their good performance in OER catalysis, as well as their low cost and abundance in the Earth’s crust [21]. Yang Shao-Horn and colleagues conducted the catalytic analysis of perovskite oxide, where they elucidated a correlation between OER activity and the occupancy of 3D electrons with e_g_ symmetry in transition metal cations on the surface. Their findings revealed that an optimized OER electrocatalyst possesses e_g_ occupancy of approximate one [22]. Liu et al. applied this principle to explain the high OER activity of CoSe_2_, which exhibits a t_2g_6e_g_1 electronic configuration, approaching the optimal e_g_ filling [23,24]. However, they overlooked the surface reconstruction or oxidation of CoSe_2_ under OER catalysis. The surface reconstruction of catalysts was frequently observed in the OER test. For example, Shahid et al. synthesized a catalyst for the OER, which consists of cobalt-containing ester salts of polyoxometalates that are immobilized on carbon nanotube fibers (Co_4_POM@CNTF). The catalyst showcases an exceptional OER performance and remarkable stability in alkaline solutions, which can be attributed to the efficient transfer of electrons and the enhanced electrochemical active surface area of the self-activating electrocatalyst that is affixed to the highly conductive carbon nanotube fibers [25]. Theoretically, the surface oxidation of CoSe_2_ would lead to the generation of Co oxides or hydroxides on the CeSe_2_ surface. This indicates that the actual catalytic phase occurs with the Co oxides/hydroxides rather than CoSe_2_. Therefore, the inner CoSe_2_ serves as an electron-transfer conductor and is also probably influenced by the electronic structure of the surface Co oxides/hydroxides. Based on the above analysis, we are inclined to believe that the good OER catalytic performance of CoSe_2_ may result from the synergistic effect of the surface oxides and the inner CoSe_2_.

NiFe-layered double-hydroxide (NiFe-LDH) and NiFeOOH species have been recognized as very efficient OER electrocatalysts both in theoretical calculations and experimental investigations [26,27]. For example, Du and coworkers prepared an OER catalyst of NiO/NiFe-LDH, which demonstrated an OER overpotential of only 205 mV at a current density of 30 mA cm^−2^, with a Tafel slope of merely 30 mV dec^−1^ in 1 M KOH solution [27].

This good catalytic performance is achieved by adjusting the adsorption energy of each intermediate, which effectively bypasses the scaling relationship. Hu et al. conducted a comprehensive investigation on FeOOH-NiOOH and NiFe LDH, unveiling the distinct electrochemical and spectroscopic characteristics between the two catalysts, despite their similar chemical compositions. Notably, in FeOOH-NiOOH, a significant portion of iron ions were observed to reside within surface γ-FeOOH clusters, contrasting the doping of iron ions within the lattice of Ni(OH)_2_/NiOOH in NiFe LDH. This discrepancy played a crucial role in the remarkable tenfold enhancement of OER activity witnessed in FeOOH-NiOOH compared to that of NiFe LDH [26].

Based on the above, we proposed to fabricate thin NiFeOOH onto a CoSe_2_ surface, aiming to achieve an optimized OER performance through the synergistic effect of NiFeOOH and CoSe_2_. To increase the accessibility of CoSe_2_, we prepared it via the selenidation of a Co-based metal–organic framework (MOF, ZIF-L) array grown on Ni foam [28,29,30]. ZIF-L is selected as the precursor because it has a thin leaf morphology and can be readily grown on nickel foam, which facilitate the following selenidation. The deposition of NiFeOOH onto CoSe_2_ was achieved through electrochemical deposition (Figure 1). The CoSe_2_@NiFeOOH/NF composite exhibited a good performance in alkaline electrolytes for the OER, demonstrating an overpotential of 254 mV at a current density of 100 mA cm^−2^ and a Tafel slope of 73 mV dec^−1^. Furthermore, it showed excellent stability with negligible current density decay after 100 h of operation. The high OER activity level can be attributed to the increased exposure of active sites and the synergistic effect of NiFeOOH outer layer and CoSe_2_ inner layer.

## 2. Experimental Section

### 2.1. Materials

Co(NO_3_)_2_·6H_2_O, 2-methylimidazole (2-MeIM, 99%), Ni(NO_3_)_2_·6H_2_O, Fe(NO_3_)_3_·9H_2_O, sodium borohydride (NaBH_4_), selenium powder (Se), and absolute ethanol were purchased from Aladdin. Ni was obtained from J&K Chemical Technology (San Jose, CA, USA) and subsequently subjected to sonication with acetone, ethanol, and deionized water for 30 min each prior to use.

### 2.2. Preparation of Cobalt-Based ZIF-L/NF

Initially, a solution of 2 × 10^−3^ mol Co (NO_3_)_2_ and another solution of 1.6 × 10^−2^ mol 2-MeIM were prepared by dissolving them individually in 40 mL of deionized water in a beaker. Then, the two solutions were combined via stirring. Subsequently, nickel foam (NF) measuring 2 × 3 cm was immersed in the mixture and left at room temperature for 4 h to generate ZIF-L/NF. To prepare it for future use, ZIF-L/NF was subjected to three rounds of rinsing with both ethanol and deionized water. Subsequently, it was dried overnight in an oven at 60 °C.

### 2.3. Synthesis of CoSe_2_/NF

The ZIF-L/NF selenidation process was carried out using a hydrothermal method. Initially, 60 mg of NaBH_4_ was completely dissolved in 25 mL of deionized water. Subsequently, 20 mg of selenium powder was added to the solution. The mixture was continuously stirred for 40 min until a light-yellow solution was formed. The solution was then transferred into a Teflon-lined stainless steel autoclave, and a ZIF-L/NF sample with dimensions of 1 cm × 1.5 cm was immersed in the solution. The autoclave was kept at a temperature of 180 °C for a duration of 8 h. Afterward, the resulting CoSe_2_/NF product was rinsed with ethanol and water and finally dried in an oven at a temperature of 60 °C for 12 h.

### 2.4. Synthesis of CoSe_2_@NiFeOOH/NF

The CoSe_2_@NiFeOOH/NF composite was synthesized via electrochemical deposition with CoSe_2_/NF as the working electrode in a solution containing Ni(NO_3_)_2_ and Fe(NO_3_)_3_ (molar ratio of Ni^2+^/Fe^3+^ = 4:1). The electrochemical deposition was conducted at a constant potential of −1.4 V vs. RHE for 8 s.

### 2.5. Characterization

X-ray power diffraction (XRD) data were recorded on a Bruker D8 ADVANCE DAVINCI. The morphologies and nanostructures were probed using field emission scanning electron microscopy (SEM, Hitachi S-4800, Japan) and transmission electron microscopy (TEM, Tecnai F20, JEM-ARM200F, USA). The surface chemical valences of the samples were analyzed using X-ray photoelectron spectroscopy (XPS) (Axis SUPRA Kratos, UK).

### 2.6. Electrocatalytic Measurement

The electrochemical measurements were conducted on a CHI760E electrochemical workstation. The as-prepared samples served as the working electrode (geometric area: 1.0 cm^2^). The counter electrode was a Pt mesh with dimensions of 1 cm × 1 cm, and an Ag/AgCl electrode was used as the reference electrode. A 1.0 M KOH electrolyte was utilized. To convert the measured potentials to the reversible hydrogen electrode (RHE) scale, the following equation was used: E_RHE_ = E_Ag/AgCl_ + 0.059 pH + 0.197 V. Linear sweep voltammetry (LSV) curves were measured at a scan rate of 5 mV s^−1^, while cyclic voltammetry (CV) curves were recorded at scan rates ranging from 10 to 50 mV s^−1^. Electrochemical impedance spectra (EIS) were measured at 1.52 V vs. RHE. All the polarization curves were IR-corrected. The double-layer capacitances were determined by analyzing the CV curves obtained at different scan rates, ranging from 10 to 50 mV s^−1^. The stability of the electrocatalyst was evaluated through chrono-amperometry performed at a constant potential. The Faraday efficiency (FE) was determined using the bubbling method following the formula: FE = 4 × F × V/(1000 × V_m_ × It), where V is the rising volume (mL) of the soap bubble over time t, I is the current density (It representing the total number of charges transferred under constant current), V_m_ is the molar volume (24.5 L mol^−1^ at 25 °C), and F is the Faraday constant (96,485 C mol^−1^).

## 3. Results and Discussion

ZIF-L was deposited onto NF prepared by facilely immersing NF into a solution of cobalt nitrate and 2-methylimidazole at room temperature. The morphology of the ZIF-L/NF sample was analyzed via SEM, which allows the detailed imaging of the sample surface at high magnification. The crystal structure of the synthesized catalyst was determined using powder XRD. Figure 2a,d,g illustrates the triangular ZIF-L plates with smooth surfaces vertically aligned on the NF. The sharp diffraction peaks at 44.5°, 51.9°, and 76.2° are attributed to the Ni substrate, and the peak at 29.4° belongs to ZIF-L. This confirms the successful fabrication of ZIF-L on NF. After selenization, ZIF-L transformed into the CoSe_2_ phase while maintaining its morphology, except that the surface became rougher (Figure 2b,e,h). Subsequently, NiFeOOH was electrochemically deposited onto CoSe_2_, resulting in a morphological change, where CoSe_2_ formed an array with increased surface roughness and additional folds. The XRD analysis of the CoSe_2_@NiFeOOH/NF sample showed no visible peaks except those of the Ni substrate, suggesting an amorphous structure in the NiFeOOH layer [31]. The absence of XRD signals for CoSe_2_ demonstrates that a relatively compact NiFeOOH layer formed on the CoSe_2_ surface. The three-dimensional CoSe_2_@NiFeOOH can aid in the transportation of electrolytes and the diffusion of reactive gases, thereby accelerating the reaction process [31].

The detailed morphology of CoSe_2_@NiFeOOH was further characterized via TEM. Figure 3a shows that CoSe_2_@NiFeOOH possesses a thin-leaf-like morphology. The weak polycrystalline rings in the selected area electron diffraction (SAED) pattern (Figure 3c) indicate the poor crystallization of these thin leaf structures in the single derived CoSe_2_@NiFeOOH sample. The structure of CoSe_2_@NiFeOOH/NF was precisely characterized using high-resolution TEM (HRTEM). On the top surface of CoSe_2_@NiFeOOH, two fringes with lattice spacings of 0.259 and 0.223 nm were observed, corresponding to the (111) and (210) planes of CoSe_2_, respectively (Figure 3b). Elemental mapping images demonstrate the homogeneous dispersion of Ni, Fe, Co, Se, and O elements throughout the entire CoSe_2_@NiFeOOH/NF, with an atomic ratio of Ni:Fe:Co:Se:O = 24:6:29:3:38 (Appendix A).

X-ray photoelectron spectroscopy (XPS) measurements were taken to analyze the elemental composition and chemical valence states of the as-prepared CoSe_2_@NiFeOOH/NF. As shown in Figure 4a, two peaks centered at 855.7 and 873.5 eV were identified as the main peaks corresponding to oxidized Ni 2p_3/2_ and Ni 2p_1/2_, respectively. Another two peaks at 861.2 and 879.3 eV were attributed to the shake-up satellite peak [32]. Figure 4b displays the Co 2p_1/2_ and Co 2p_3/2_ peaks, which can be further divided into four peaks located at 796.8/780.8 eV (Co^2+^) and 783.8/802.2 eV (Co^3+^). This suggests that the Co atom in CoSe_2_@NiFeOOH/NF is predominantly in the valence states of +2 and +3 [33]. Due to the overlap of the 2p peak of Fe with the Auger peak of Co, it is impossible to deconvolute the Fe 2p XPS peak. Nevertheless, we can determine the presence of Fe and its oxidation state from the two peaks in the XPS raw data of Fe 2p. In the XPS spectra of Fe 2p (Figure 4c), two characteristic peaks at 711.1 and 724.6 eV correspond to Fe 2p_3/2_ and Fe 2p_1/2_, respectively, indicating the presence of Fe^3+^ [33]. The findings collectively indicate that Ni and Fe exist in Ni and Fe oxidation states in CoSe_2_@NiFeOOH/NF. In addition, as shown in Figure 4d, Se 3d_5/2_ (54.7 eV) and Se 3d_3/2_ (55.4 eV) confirm the existence of selenium–metal bonds, while the peak at 59.2 eV corresponds to a Se-O bond, which is due to the surface oxidation in the air environment [34,35,36,37].

The catalysts’ OER performance was assessed using an electrochemical three-electrode system in 1 M KOH alkaline solution. As shown in Figure 5a, CoSe_2_@NiFeOOH/NF exhibits an overpotential of 254 mV at 100 mA cm^−2^, which is significantly lower than those of CoSe_2_/NF, NiFe-LDH/NF, NF/Selenization, CoSe_2_@NiOOH/NF, and CoSe_2_@FeOOH/NF. The OER performance of CoSe_2_@NiFeOOH/NF is comparable to those of other reported OER electrocatalysts, such as NiSe_2_-CoSe_2_ [38], SiO_2_/Co_x_P [39], and Ni(OH)_2_@NiS_2_ [40] (Appendix A). The Tafel slopes of the four samples, namely CoSe_2_@FeOOH/NF, CoSe_2_@NiOOH/NF, CoSe_2_/NF, and CoSe_2_@NiFeOOH/NF, were calculated to analyze their kinetics. As shown in Figure 5c, CoSe_2_@NiFeOOH possesses the smallest Tafel slope of 73 mV dec^−1^, which is much smaller than those of CoSe_2_@FeOOH/NF (102 mV dec^−1^), CoSe_2_/NF (119 mV dec^−1^), and CoSe_2_@NiOOH/NF (126 mV dec^−1^), suggesting faster OER kinetics. To gain further insight into the kinetics of electron transfer, EIS measurements were taken. Figure 5b displays the Nyquist plots of all the samples. Notably, the CoSe_2_@NiFeOOH/NF sample exhibits the smallest diameter of the semicircle in the high frequency region, which unequivocally confirms the presence of charge transfer resistance (R_ct_). This resistance facilitates smooth charge transfer, ultimately resulting in excellent oxygen evolution reaction (OER) activity. Additionally, to understand the intrinsic catalytic activity of CoSe_2_@NiFeOOH/NF for the OER, the electrochemical surface area (ECSA) was estimated by measuring the double-layer capacitance (C_dl_) in the potential range of 1.07–1.17 V at scan rates ranging from 10 to 50 mV s^−1^ (Appendix A). Interestingly, CoSe_2_@NiFeOOH/NF exhibits the highest Cdl value of 29.0 mF cm^−2^ among all the samples, surpassing those of CoSe_2_@NiOOH/NF (23.1 mF cm^−2^) and CoSe_2_@FeOOH/NF (20.5 mF cm^−2^). Furthermore, the electrocatalytic stability of CoSe_2_@NiFeOOH/NF was evaluated. As depicted in Figure 5h, the current variation is negligible after a 100 h continuous test at 1.54 V. Furthermore, we also probed the morphology and electronic structural characterization of CoSe_2_@NiFeOOH/NF after a 100 h stability test. As shown in Appendix A, no visible variation in the XPS spectra and morphology of CoSe_2_@NiFeOOH/NF after the stability test was observed, demonstrating excellent stability.

The overpotential increase for every 50 mA cm^−2^ of current density is illustrated in Figure 5f. It is evident from Figure 5a that CoSe_2_@NiFeOOH/NF has a significantly lower overpotential than those of CoSe_2_@FeOOH/NF and CoSe_2_@NiOOH/NF, signifying the higher catalytic activity level of CoSe_2_@NiFeOOH. Figure 5e illustrates the chrono-potentiometric curve of CoSe_2_@NiFeOOH, showing multiple steps with the current density incrementing from 10 to 100 mA cm^−2^ in ten steps. In the initial step, the potential rapidly stabilizes at 1.44 V and remains nearly constant for a duration of 320 s. Similar phenomena are observed in subsequent steps, indicating the exceptional mass transport and electronic conductivity characteristics of CoSe_2_@NiFeOOH/NF. The Faradaic efficiency of the OER with the CoSe_2_@NiFeOOH/NF electrodes was evaluated by measuring the O_2_ produced during a constant current test experiment. According to Figure 5g, the measured amount of O_2_ is in close agreement with the theoretical yield, resulting in a remarkable Faradaic efficiency of 97.8% for the CoSe_2_@NiFeOOH/NF electrode.

## 4. Conclusions

In summary, a three-dimensional CoSe_2_@NiFeOOH sample supported on nickel foam for efficient OER catalysis was fabricated. The excellent conductivity of CoSe_2_ and the synergistic effect between CoSe_2_ and NiFeOOH facilitate OER catalysis on CoSe_2_@NiFeOOH/NF. The CoSe_2_@NiFeOOH/NF catalyst demonstrated an excellent performance in the alkaline electrolyte, achieving an overpotential of 254 mV at 100 mA cm^−2^ and a Tafel slope of 73 mV dec^−1^. Furthermore, the catalyst demonstrated remarkable stability with minimal decay in the current density, even after 100 h. The findings of this study represent a prototype of a synergistic strategy for creating highly efficient electrocatalysts.

## Figures and Tables

**Figure 1 nanomaterials-13-02621-f001:**
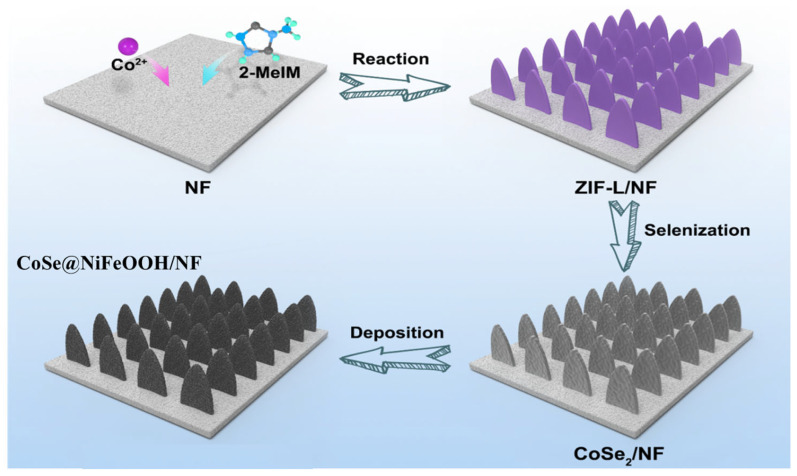
Schematic illustration of the routes to CoSe_2_@NiFeOOH/NF.

**Figure 2 nanomaterials-13-02621-f002:**
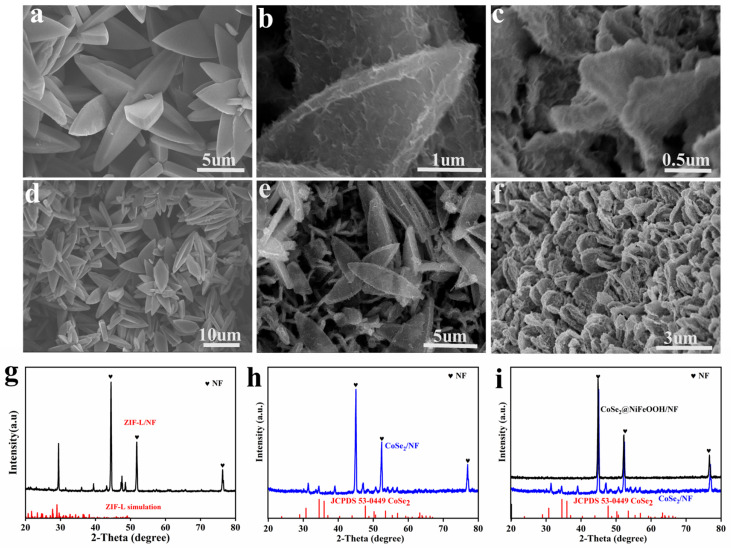
SEM images of (**a**,**d**) ZIF-L, (**b**,**e**) CoSe_2_, and (**c**,**f**) CoSe_2_@NiFeOOH/NF. XRD patterns of (**g**) ZIF-L/NF, (**h**) CoSe_2_/NF, and (**i**) CoSe_2_@NiFeOOH/NF.

**Figure 3 nanomaterials-13-02621-f003:**
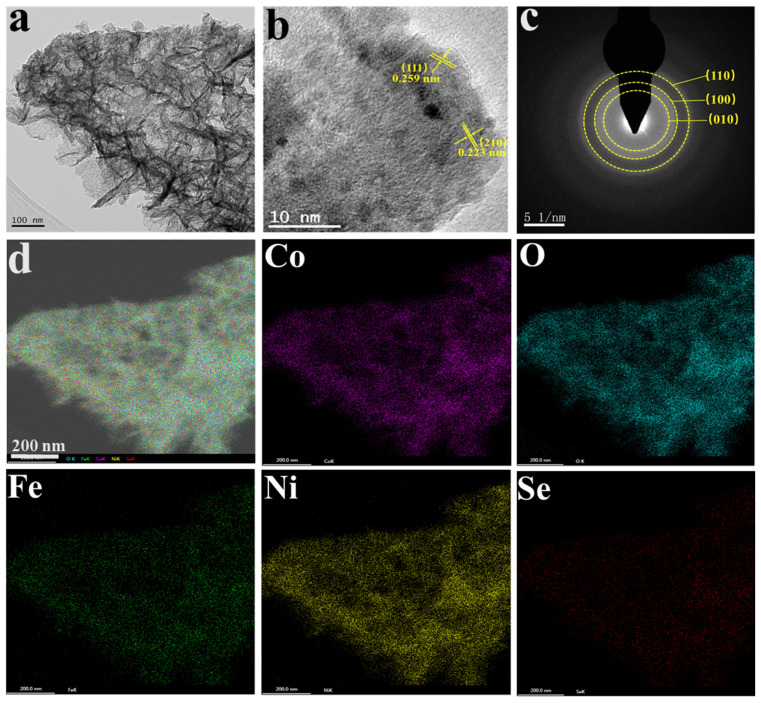
(**a**) TEM, (**b**) HRTEM, (**c**) SAED pattern, and (**d**) STEM images and the corresponding elemental mapping images of Co, O, Fe, Ni, and Se of CoSe_2_@NiFeOOH/NF.

**Figure 4 nanomaterials-13-02621-f004:**
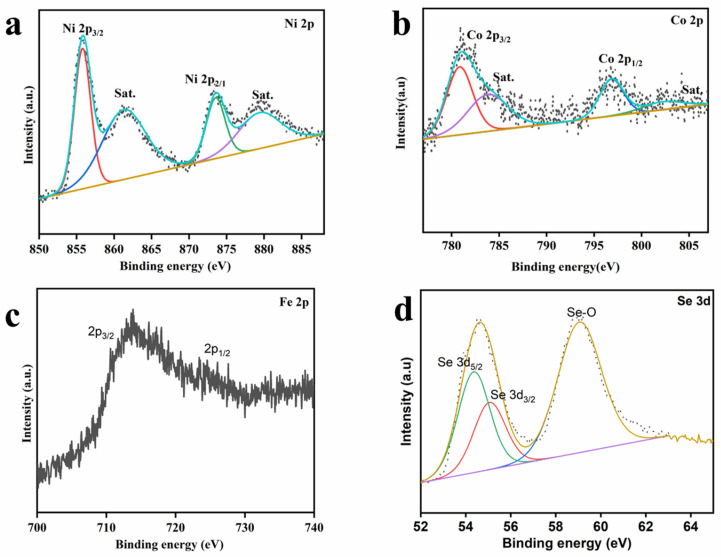
High-resolution XPS spectra of CoSe_2_@NiFeOOH/NF (**a**) Ni, (**b**) Co, (**c**) Fe, (**d**) Se.

**Figure 5 nanomaterials-13-02621-f005:**
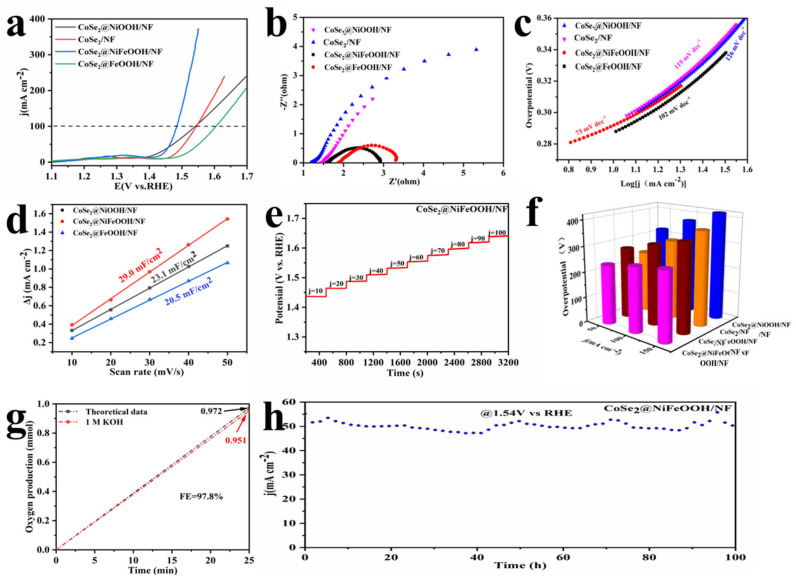
(**a**) LSV curves for CoSe_2_/NF, CoSe_2_@FeOOH/NF, CoSe_2_@NiOOH/NF, NiFe-LDH/NF, NF/selenization, and CoSe_2_@NiFeOOH/NF. (**b**) EIS curves, (**c**) Tafel plots, and (**d**) C_dl_ plots. (**e**) Rate capability evaluation of CoSe_2_@NiFeOOH/NF. (**f**) Catalyst overpotentials at different current densities. (**g**) Faraday efficiency. (**h**) i–t curve for CoSe_2_@NiFeOOH at the η = 0.31 V for 100 h.

## Data Availability

We would like to share our data upon request from the authors.

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
