# Peer review of "MOF-Derived CoSe2@NiFeOOH Arrays for Efficient Oxygen Evolution Reaction"

_nanomaterials, 2023, doi:10.3390/nano13192621_

Round 1

Reviewer 1 Report

The authors have reported a novel MOF-derived CoSe2@NiFeOOH porous arrays material for the oxygen evolution reaction (OER). Given the well-established material characterization and efficient catalytic performance, I recommend accepting it for publication in the Nanomaterials journal after major revisions. The following issues need clarification:

1.     In the synthesis of the materials, the authors involve complex components such as MOF precursors, CoSe2, NiFeOOH, etc. However, there is minimal introduction or review of the application of these materials in the oxygen evolution reaction (OER). This lack of context hinders readers' understanding of the progress in related research. Therefore, I suggest that the authors provide a comprehensive review in the Introduction section, covering the performance of these materials in the field of OER and common strategies employed to enhance their performance.

2.     Why did the authors choose ZIF-L as the MOF precursor for material synthesis? Considering the diverse types and structures of MOFs available, it is crucial to justify this choice. There are numerous MOF types and structures, such as those discussed in "Recent advances in the synthesis of nanoscale hierarchically porous metal–organic frameworks" and "Synthesis strategies of metal-organic frameworks for CO2 capture." These sources introduce various MOF materials and structures. Do materials like CoSe2, synthesized using similar methods, offer advantages in catalysis? Please provide a rationale for your choice.

3.     The synthesis conditions for the materials appear to be fixed. How did the authors determine these synthesis conditions? If the synthesis conditions were modified to change the ratio of active components, how would it affect the catalytic performance?

4.     Please specify the elemental composition of CoSe2@NiFeOOH porous arrays materials.

5.     The XPS peak fitting for Fe and Se needs correction. For Fe 2p, it should include 2p3/2 and 2p1/2 along with their satellite peaks, as demonstrated in the article titled "Bioinspired Nanocomposites with Self-Adaptive Stress Dispersion for Super-Foldable Electrodes." Additionally, Se fitting should encompass both 3d3/2 and 3d5/2 peaks within the energy range of 52-58, as supported by relevant literature. Please re-fit the XPS data according to these recommendations.

6.     In the section comparing the catalytic performance, consider including comparisons with similar or closely related materials, such as "NiSe2-CoSe2 with a Hybrid Nanorods and Nanoparticles Structure for Efficient Oxygen Evolution Reaction," "Coupling of ultrasmall and small CoxP nanoparticles confined in a porous SiO2 matrix for a robust oxygen evolution reaction," and "Ni(OH)2 Derived from NiS2 Induced by Reflux Playing Three Roles for Hydrogen/Oxygen Evolution Reaction."

7.     Catalytic performance is closely related to the material's specific surface area. Please provide the specific surface area of the material in your manuscript.

It's fine. Minor editing of English language required

Author Response

Dear editor,

We are very grateful to your and the reviewers’ critical comments and thoughtful suggestions. Based on these comments and suggestions, we have made careful revision on the original manuscript entitled " MOF-derived CoSe2@NiFeOOH porous arrays for efficient oxygen evolution reaction" (Manuscript ID: nanomaterials-2606723). We responded point by point to the comments of reviewer #1 as listed below, along with a clear indication of the location of the revision.

In specific response to the points raised by reviewer #1:

Q1: In the synthesis of the materials, the authors involve complex components such as MOF precursors, CoSe2, NiFeOOH, etc. However, there is minimal introduction or review of the application of these materials in the oxygen evolution reaction (OER). This lack of context hinders readers' understanding of the progress in related research. Therefore, I suggest that the authors provide a comprehensive review in the Introduction section, covering the performance of these materials in the field of OER and common strategies employed to enhance their performance.

Our response: Thank you for your suggestion. We have enhanced the introduction with recent literatures.

Q2: Why did the authors choose ZIF-L as the MOF precursor for material synthesis? Considering the diverse types and structures of MOFs available, it is crucial to justify this choice. There are numerous MOF types and structures, such as those discussed in "Recent advances in the synthesis of nanoscale hierarchically porous metal–organic frameworks" and "Synthesis strategies of metal-organic frameworks for CO2 capture." These sources introduce various MOF materials and structures. Do materials like CoSe2, synthesized using similar methods, offer advantages in catalysis? Please provide a rationale for your choice.

Our response: Thanks for your comment. The reasons for the selection of ZIF is: 1) ZIF-L is a co-based MOF which can be ready synthesized and defined; 2) the thin-leaf morphology of ZIL-L grown on nickel foam facilitate the selenidation. We have added the related discussion in the manuscript.

Q3: The synthesis conditions for the materials appear to be fixed. How did the authors determine these synthesis conditions? If the synthesis conditions were modified to change the ratio of active components, how would it affect the catalytic performance?

Our response: Thank you for your comments. The selenization condition is based on our previous study (Inorg. Chem. 2022, 61, 19031−19038), and the synthesis conditions for NiFeOOH electrodeposition are based on literatures reported (J. Mater. Chem. A, 2015, 3, 6921–6928). Since these conditions were optimized in our previous study or literatures, we did not investigate the effects on ratio of active components.

Q4: Please specify the elemental composition of CoSe2@NiFeOOH porous arrays materials.

Our response: Thanks for your comment. The EDS element analysis of CoSe2@NiFeOOH is provided. The mole ratio of Co, Ni, Fe, O and Se is 29:24:6:3:38. We have added the related discussion.

Q5: The XPS peak fitting for Fe and Se needs correction. For Fe 2p, it should include 2p3/2 and 2p1/2 along with their satellite peaks, as demonstrated in the article titled "Bioinspired Nanocomposites with Self-Adaptive Stress Dispersion for Super-Foldable Electrodes." Additionally, Se fitting should encompass both 3d3/2 and 3d5/2 peaks within the energy range of 52-58 eV, as supported by relevant literature. Please refit the XPS data according to these recommendations.

Our response: Thank you for your suggestion. Due to the overlap of the 2p peak of Fe with the Auger peak of Co, it is impossible to deconvolute the Fe 2p XPS peak. Nevertheless, we can determine the presence of Fe and its oxidation state from the two peaks in the XPS raw data of Fe 2p. We have refitted the XPS 3d spectrum of Se according to literatures. We have added the related discussion in the manuscript.

Q6: In the section comparing the catalytic performance, consider including comparisons with similar or closely related materials, such as "NiSe2-CoSe2 with a Hybrid Nanorods and Nanoparticles Structure for Efficient Oxygen Evolution Reaction," "Coupling of ultrasmall and small CoxP nanoparticles confined in a porous SiO2 matrix for a robust oxygen evolution reaction," and "Ni(OH)2 Derived from NiS2 Induced by Reflux Playing Three Roles for Hydrogen/Oxygen Evolution Reaction."

Our response: Thank you for your suggestion. We have added the references for comparisons.

Q7: Catalytic performance is closely related to the material's specific surface area. Please provide the specific surface area of the material in your manuscript.

Our response: Thank you for your suggestion. We have conducted N2 sorption measurements according to the literatures. It is shown that CoSe2@NiFeOOH arrays are nonporous. We appreciate the concern of the reviewer. We have deleted the “porous” statement in the manuscript. Since N2 sorption measurements can only precisely detect the micro and mesopores, the abundant gaps between CoSe2@NiFeOOH nanoplates as observed by SEM cannot characterized by the N2 sorption measurement. However, these gaps can efficiently facilitate the mass transfer and increase the exposure of active sites. We thus estimate the electrochemical surface area (ECSA) by measuring the double-layer capacitance (Cdl) in the potential range of 1.07-1.17 V at scan rates ranging from 10 to 50 mV s-1 (Figure S1). Interestingly, CoSe2@NiFeOOH/NF exhibits the highest Cdl value of 29.0 mF cm-2 among all the samples.

We appreciate for your warm work earnestly, and hope that the correction will meet with approval. The manuscript has been overall checked, and the changes marked in blue. We hope that these revisions are sufficient to make our manuscript acceptable for publication in Nanomaterials. If you believe that any additional clarifications need to be addressed, I will be happy to include them. Once again, thank you very much for your comments and suggestions.

Yours sincerely,
Yunan Wang
Ningbo Institute of Materials Technology and Engineering, Chinese Academy of Sciences
E-mail: [email protected]

Reviewer 2 Report

This work presents MOF-derived CoSe2@NiFeOOH porous arrays for efficient oxygen evolution reaction. Authors performed various physical characterizations to discuss the findings and high performance is attributed to synergistic interaction between CoSe2 and NiFeOOH. This manuscript is poorly managed, and all the figure captions are incorrect with respect to the text. Therefore, I do not find the article suitable for publication in Nanomaterials. Some additional comments are;

1. The introduction is lacking sufficient background, Authors are suggested to enrich the introduction with recent literature.

2. Fig, 4c, the Fe XPS is not deconvoluted. why authors use raw XPS peak when it doesn't give any useful information about electronic structure.

3. Authors are suggested to revise the manuscript carefully, there are huge blunders in figure numbering and captions with respect to text. Please double-check the Figure captions with the text, mostly the figures are described wrongly in the text, for example, Figure 2 is discussed as, Fig 1a, Figure 2b.... similarly, Figure 4 is mentioned as Figure 3. 

4. Catalyst stability is a critical factor. Also, the surface reconstruction and modification is usual phenomenon in OER catalysts. Thus a post durability morphology and electronic strucutral characterization is improtant to understand the practical applications of the catalyst. Authors are suggested to perform a potential cycling test and provide post-durability morphology and electronic structural analysis. 

5. References should be updated with recent literature, some recomendations are, 10.1021/acsami.2c20246, 10.3390/catal12101242, 10.1016/j.trechm.2022.07.007

NA

Author Response

Dear editor,

We are very grateful to your and the reviewers’ critical comments and thoughtful suggestions. Based on these comments and suggestions, we have made careful revision on the original manuscript entitled " MOF-derived CoSe2@NiFeOOH porous arrays for efficient oxygen evolution reaction" (Manuscript ID: nanomaterials-2606723). We responded point by point to the comments of reviewer #2 as listed below, along with a clear indication of the location of the revision.

In specific response to the points raised by reviewer #2:

Q1. The introduction is lacking sufficient background, Authors are suggested to enrich the introduction with recent literature.

Our response: We have enriched the introduction with recent literatures.

Q2. Fig. 4c, the Fe XPS is not deconvoluted. why authors use raw XPS peak when it doesn't give any useful information about electronic structure.

Our response: Due to the overlap of the 2p peak of Fe with the Auger peak of Co, it is not possible to correctly deconvolute the Fe 2p XPS. Nevertheless, we can confirm the presence of iron and its oxidation state from the two peaks in the XPS raw data of Fe 2p.

Q3. Authors are suggested to revise the manuscript carefully, there are huge blunders in figure numbering and captions with respect to text. Please double-check the Figure captions with the text, mostly the figures are described wrongly in the text, for example, Figure 2 is discussed as, Fig 1a, Figure 2b.... similarly, Figure 4 is mentioned as Figure 3.

Our response: We are very sorry for our negligence. We have now carefully revised the manuscript.

Q4. Catalyst stability is a critical factor. Also, the surface reconstruction and modification is usual phenomenon in OER catalysts. Thus, a post durability morphology and electronic structural characterization is important to understand the practical applications of the catalyst. Authors are suggested to perform a potential cycling test and provide post-durability morphology and electronic structural analysis.

Our response: We have provided TEM images and XPS spectra after 100 h durability test in Figure S2 and S3. From the TEM images and XPS spectra, it can be observed that there is no notable change in the structure of the catalyst. We have added the related discussion in the manuscript.

Q5. References should be updated with recent literature, some recomendations are, 10.1021/acsami.2c20246, 10.3390/catal12101242, 10.1016/j.trechm.2022.07.007

Our response: Thanks for your suggestion, we have added the suggested literatures.

We appreciate for your warm work earnestly, and hope that the correction will meet with approval. The manuscript has been overall checked, and the changes marked in blue. We hope that these revisions are sufficient to make our manuscript acceptable for publication in Nanomaterials. If you believe that any additional clarifications need to be addressed, I will be happy to include them. Once again, thank you very much for your comments and suggestions.

Yours sincerely,
Yunan Wang
Ningbo Institute of Materials Technology and Engineering, Chinese Academy of Sciences
E-mail: [email protected]

Reviewer 3 Report

            This manuscript reports to MOF-derived CoSe2@NiFeOOH porous arrays for efficient oxygen evolution reaction works are interesting. Moreover, in this study, a CoSe2@NiFeOOH/NF catalyst demonstrates outstanding activity for the OER in an alkaline electrolyte. It exhibits a low over-potential (η) of 254 mV at 100 mA cm−2, a small Tafel slope of 73 mV dec−1, and excellent high stability. The high performance of CoSe2@NiFeOOH/NF can be attributed to the combination of the high conductivity of the inner layer and the synergistic effect between CoSe2 and NiFeOOH. The overall scientific organization of this article is satisfactory with some major revisions as must be needed before acceptance:

Reviewer comments and suggestions:                                    

1.      The author does provide, what is the novelty of the present work with discusses the recent related materials in the introduction part.

2.      What is the advantage of this catalyst? why did the authors select this NiFeOOH catalyst?

3.      Consistently express the units used in the manuscript. Please keep the Figure. * or Fig.* style consistent throughout the main content. Also, make sure that the Figure axis units are correct.

4.      I suggested that the author may provide the morphology properties and FT-IR spectra or XPS after 100h of stability test samples. Since it may confirm the structural stability of tested samples.

5.      Some typical errors have been obtained in this manuscript. The manuscript needs extensive editing and the quality of the paper improved substantially.

6.      An author should provide more information about a possible mechanism to improve the manuscript's brightness.

7.      What is the synergistic interaction between CoSe2 to NiFeOOH? What's the mechanism behind it?

Author Response

Dear editor,

We are very grateful to your and the reviewers’ critical comments and thoughtful suggestions. Based on these comments and suggestions, we have made careful revision on the original manuscript entitled " MOF-derived CoSe2@NiFeOOH porous arrays for efficient oxygen evolution reaction" (Manuscript ID: nanomaterials-2606723). We responded point by point to the comments of reviewer #3 as listed below, along with a clear indication of the location of the revision.

In specific response to the points raised by reviewer #3:

Q1. The author does provide, what is the novelty of the present work with discusses the recent related materials in the introduction part.

Our response: Thanks for the comment.

Q2. What is the advantage of this catalyst? why did the authors select this NiFeOOH catalyst?

Our response: Thanks for the comment. Since NiFeOOH species have been recognized as very efficient OER electrocatalysts both from the theoretical calculations and experimental investigations, we proposed to fabricate thin NiFeOOH onto CoSe2 surface which has outstanding electron conductivity and high OER activity. the inner CoSe2 serves as an electron-transfer conductor, and is also probably influence the electronic structure of the surface Co oxides/hydroxides.

Q3. Consistently express the units used in the manuscript. Please keep the Figure. or Fig. style consistent throughout the main content. Also, make sure that the Figure axis units are correct.

Our response: We have carefully examined and corrected it.

Q4. Suggested that the author may provide the morphology properties and FT-IR spectra or XPS after 100-h of stability test samples. Since it may confirm the structural stability of tested samples.

Our response: Thanks for the suggestions, we have provided XPS spectra and TEM image after 100-h of stability test samples in Figure S2 and 3. From the TEM images and XPS spectra, it can be observed that there is no notable change in the structure of the catalyst.

Q5. Some typical errors have been obtained in this manuscript. The manuscript needs extensive editing and the quality of the paper improved substantially.

Our response: We have carefully revised the manuscript.

Q6 and Q7. Author should provide more information about a possible mechanism to improve the manuscript's. What is the synergistic interaction between CoSe2 to NiFeOOH? What's the mechanism behind it?

Our response: We admit it is hard to clearly demonstrate the synergistic interaction between CoSe2 to NiFeOOH. However, it is reasonable to claim that NiFeOOH layer serves as the catalytic sites and protect most of the inner CoSe2 layer, while CoSe2 with outstanding electron conductivity serves as the electron-transfer conductor. In addition, the partially reconstruction of CoSe2 may influence the electronic structure of the surface Co oxides/hydroxides.

We appreciate for your warm work earnestly, and hope that the correction will meet with approval. The manuscript has been overall checked, and the changes marked in blue. We hope that these revisions are sufficient to make our manuscript acceptable for publication in Nanomaterials. If you believe that any additional clarifications need to be addressed, I will be happy to include them. Once again, thank you very much for your comments and suggestions.

Yours sincerely,
Yunan Wang
Ningbo Institute of Materials Technology and Engineering, Chinese Academy of Sciences
E-mail: [email protected]

Reviewer 4 Report

In the present work, a useful procedure for the synthesis of CoSe2@NiFeOOH arrays was presented. The as-prepared composites were found to be active and stable catalysts for OER reactions. Both the design and implementation of this research deserve to be published. However, there is only one concern.
The authors have stated that their systems are porous. However, there is no direct evidence to support this presumption. To support this assumption, N2 sorption measurements must be made. These data can be used to decide whether the as-prepared structures are indeed porous. To explain the present data, I suggest using the following publication: S. Murath et al, “Morphological aspects determine the catalytic activity of porous hydrocalumites: the role of the sacrificial templates” Mater Today Chem. 23 (2023) 100682. DOI: https://doi.org/10.1016/j.mtchem.2021.100682

Considering the above-mentioned, I suggest this manuscript for publication in an MDPI journal “Nanomaterials” after minor revision.

Author Response

Dear editor,

We are very grateful to your and the reviewers’ critical comments and thoughtful suggestions. Based on these comments and suggestions, we have made careful revision on the original manuscript entitled " MOF-derived CoSe2@NiFeOOH porous arrays for efficient oxygen evolution reaction" (Manuscript ID: nanomaterials-2606723). We responded point by point to the comments of reviewer #4 as listed below, along with a clear indication of the location of the revision.

In specific response to the points raised by reviewer #4:

In the present work, a useful procedure for the synthesis of CoSe2@NiFeOOH arrays was presented. The as-prepared composites were found to be active and stable catalysts for OER reactions. Both the design and implementation of this research deserve to be published. However, there is only one concern. The authors have stated that their systems are porous. However, there is no direct evidence to support this presumption. To support this assumption, N2 sorption measurements must be made. These data can be used to decide whether the as-prepared structures are indeed porous. To explain the present data, I suggest using the following publication: S. Murath et al, “Morphological aspects determine the catalytic activity of porous hydrocalumites: the role of the sacrificial templates” Mater Today Chem. 23 (2023) 100682. DOI: https://doi.org/10.1016/j.mtchem.2021.100682. Considering the above-mentioned, I suggest this manuscript for publication in an MDPI journal “Nanomaterials” after minor revision.

Our response: Thank you for your suggestion. We have conducted N2 sorption measurements according to the literatures. It is shown that CoSe2@NiFeOOH arrays are nonporous. We appreciate the concern of the reviewer. We have deleted the “porous” statement in the manuscript.

We appreciate for your warm work earnestly, and hope that the correction will meet with approval. The manuscript has been overall checked, and the changes marked in blue. We hope that these revisions are sufficient to make our manuscript acceptable for publication in Nanomaterials. If you believe that any additional clarifications need to be addressed, I will be happy to include them. Once again, thank you very much for your comments and suggestions.

Yours sincerely,
Yunan Wang
Ningbo Institute of Materials Technology and Engineering, Chinese Academy of Sciences
E-mail: [email protected]

Round 2

Reviewer 1 Report

The issue has been resolved after modifications, and the quality of the manuscript has been effectively improved. Therefore, I agree to its publication in the Nanomaterials journal.

Reviewer 2 Report

To be accepted 

N/A